# NaBiS$_2$ as a Novel Indirect Bandgap Full Spectrum Photocatalyst: Synthesis and Application

**Huanchun Wang** [1,2,*] , **Zheng Xie** [2,*] , **Xuanjun Wang** [2,3] **and Ying Jia** [2]

[1]   Shaanxi Engineering Laboratory for Advanced Energy Technology,
     School of Materials Science & Engineering, Shaanxi Normal University, Xi'an 710119, China
[2]   High-Tech Institute of Xi'an, Xi'an 710025, China
[3]   Shaanxi Key Laboratory of Special Fuel Chemistry and Material, Xi'an 710025, China
[*]   Correspondence: wang-hc12@tsinghua.org.cn (H.W.); xiezheng1003@163.com (Z.X.)

**Abstract:** Photocatalysts with a superior activity range, from ultraviolet (UV) to near-infrared (NIR) light, are attractive for solar utilization. From this perspective, sulfides are promising due to their narrower bandgap than oxides. In this report, NaBiS$_2$ was synthesized hydrothermally under mild conditions by adjusting the alkaline amount. The rough NaBiS$_2$ nanosheets possessed various surface atomic configurations on their surfaces, including amorphous clusters and amorphous nano-domains, revealed by HRTEM. A theoretical investigation of the band structure employing the density functional theory (DFT) method for the first time indicated that NaBiS$_2$ is an indirect bandgap semiconductor with a narrow bandgap of 1.02 eV. Experimentally, it showed excellent photocatalytic activity for the degradation of methyl blue under UV, visible light and NIR light due to its experimental bandgap width of 1.32 eV. A degradation rate of 99.6% was reached after 80 min under full spectrum irradiation.

**Keywords:** hydrothermal; NaBiS$_2$; broadband spectrum; indirect bandgap semiconductor

## 1. Introduction

Though massive efforts have been made to conquer the predicament of environmental pollution and the energy crisis, it is still an arduous task to completely settle this dilemma worldwide. The utilization of solar energy in photocatalytic [1–3] and photovoltaic [4–6] manners is considered a promising strategy for green and sustainable development. The photocatalytic degradation of hazardous substances based on semiconductor catalysts could thoroughly decompose organic pollutants into inorganic molecules theoretically, and possesses the superiorities of low energy investment and being environmentally friendly. However, most research focuses on wide energy bandgap materials such as TiO$_2$ [7], Bi$_2$WO$_4$ [8,9] and g-C$_3$N$_4$ [10,11], which only absorb ultraviolet (UV) and visible light. Recently, semiconductors with a narrow bandgap ($E_g$ < 2.0 eV) have aroused extensive concern [12–16].

Unlike the classical UV light utilization mechanism, various mechanisms are involved for full solar light spectrum utilization [13,17] and the exploitation of broadband spectrum photocatalysts is still one of the most promising. The width of the bandgap is completely dependent on the composition of the valence band and the conduction band. Compared to oxides, sulfides have a shallower energy level and usually possess a narrower bandgap. This is beneficial for the development of near-infrared (NIR) active photocatalysts. Based on this assumption, a handful of materials with NIR photocatalytic activity were studied recently. An original WS$_2$ ($E_g$ = 1.35 eV) nanosheet, fabricated by pyrolysis of (NH$_4$)WS$_4$ in a hydrogen flow, was investigated as a full solar light spectrum photocatalyst for methyl orange (MO) photodegradation in various pH conditions [13]. As reported by the same group, β-In$_2$S$_3$ demonstrated an excellent broadband photocatalytic activity range, from ultraviolet to near-infrared, in degrading MO [15]. Recently, Sb$_2$S$_3$ with an $E_g$ of 1.48 eV was deemed to have remarkable near-infrared-driven

Cr(VI) reduction conversion efficiency in an aqueous solution, and this novel property could be further improved by being composited with $MoS_2$ [18]. Additionally, by means of compositing with carbon or other materials, several binary sulfides revealed non-ignorable photocatalytic activity under full solar spectrum irradiation, though these sulfides showed no NIR photocatalytic activity intrinsically [19–22]. On account of their air-sensibility, those sulfides were usually fabricated through a rigorous process, which was an obstacle to their practical application for solar energy utilization.

Herein, we report a facile hydrothermal method for ternary sulfide $NaBiS_2$ synthesis as a novel broadband spectrum photocatalyst, without a rigorous process or poisonous reagents. The alkaline amount is believed to be pivotal for $NaBiS_2$ formation. The band structure was investigated theoretically by the density functional theory (DFT) method using a Vienna ab-initio simulation package (VASP), and $NaBiS_2$ was deemed to be an indirect bandgap semiconductor. Its superior broadband spectrum photocatalytic activity and chemical stability were confirmed through a degradation test of methyl blue (MB). Both the experimental and theoretical results indicated the prospect of the application of $NaBiS_2$ as a novel photocatalyst, and provided a new opportunity for expanding the utilization of solar light.

## 2. Results and Discussion

Figure 1a shows the XRD pattern of the sample synthesized through a typical hydrothermal procedure at 160 °C. All of the diffraction peaks can be indexed to $NaBiS_2$ without any impurity by careful comparison with JCPDS card file no. 08-0406, and agree well with NaCl-type structure. The sharp peaks indicate well-developed crystallinity and the grain size, evaluated by the Scherer equation, is 65.4 nm.

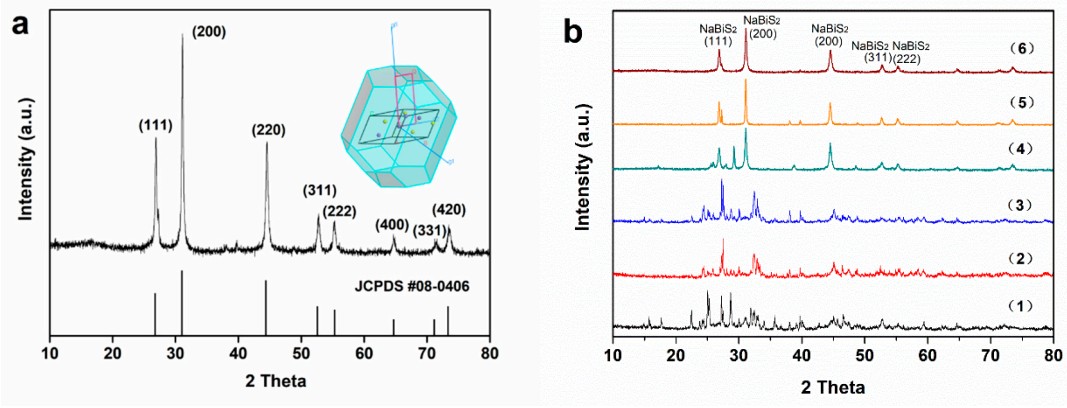

**Figure 1.** XRD patterns of $NaBiS_2$. (**a**) a typical XRD result of $NaBiS_2$; (**b**) XRD results of samples fabricated at various mole ratios of $Bi(NO_3):Na_2S:NaOH$, (1)1:2:2; (2)1:5:2; (3)1:5:5; (4)1:2:10; (5)1:5:10; (6)1:10:10.

$NaBiS_2$ was reported to be synthesized from mixtures of $Bi(NO_3)_3$ and L-cysteine [23], or by a two-step hydrothermal process from $Bi(NO_3)_3 \cdot 5H_2O$ and $Na_2S \cdot 9H_2O$ [24], both of which were reacted in NaOH aqueous solution. Nevertheless, $NaBiS_2$ was believed not to be the most stable product in the hydrothermal process. The initially formed polycrystals would decompose in the following reaction and tended to transform into $Bi_2S_3$ [25,26]. A contradictory phenomenon indicated that the formation mechanism of $NaBiS_2$ is still unclear, and an in-depth study on the conditions for $NaBiS_2$ synthesis is necessary.

With an excess amount of $Na_2S$ for $NaBiS_2$ stoichiometry and concentrated alkaline solutions provided by NaOH, $NaBiS_2$ was obtained through the typical procedure. Fabrications with less $Na_2S$ or NaOH were also carried out, respectively. As shown in Figure 1b, the formation of $NaBiS_2$ was strongly correlated to the amount of NaOH. Without plenty of NaOH in the solution (the mole ratio of $Bi(NO_3)_3$ to NaOH was less than 1:5), no $NaBiS_2$ was detected by XRD, even if the amount of $S^{2-}$ was excessive (samples 1–3 in Figure 1b). The promiscuous diffraction peaks could be indexed to $BiONO_3$,

$Bi_2O_3$ or $Bi_2S_3$, which were hydrolysis products or the sulfide precipitation of $Bi(NO_3)_3$. Under a high alkalinity condition, crystalline $NaBiS_2$ could be obtained easily even with a stoichiometric ratio of $S^{2-}$ (samples 4–6) and impurities were completely suppressed when $Na_2S$ was in excess (sample 6). An alkaline condition was found to be beneficial and crucial to the formation of crystalline $NaBiS_2$. This might be due to the increased solubility of the hydrolysis product of Bi ions under the circumstance of an alkaline solution [27,28]. In this work, a mole ratio of 1:10 for $Bi(NO_3)_3$ to NaOH was found to be the critical concentration for $NaBiS_2$ hydrothermal synthesis in the above mentioned procedure.

Morphologies were checked by employing SEM and TEM. As shown in Figure 2a, the as-fabricated sediments were nanoplates with a thickness less than 100 nm. These nanostructures were irregular in shape and their surfaces were rough. Obvious grain boundaries can be observed in a low magnified TEM image (Figure 2b), and the inhomogeneous contrast indicates the polycrystallinity of the as-prepared sheets. A high magnified TEM image (Figure 2c) reveals that these grain boundaries may be derived from the incomplete development of the crystal lattice. Several besieged nanoscale surface domains show different atomic configurations compared with the surface atomic arrangement of the matrix. For example, region 1 in Figure 1c shows more incompact surface atom arrangement than that of the matrix. The latter has the lattice distance of 0.4263 nm in [010] direction and 0.3612 nm in [1$\bar{1}$0] direction. As for region 3, though the atom arrangement is still uniform, the edge of it is amorphous. More amorphous nano-domains are shown in region 2 and region 4. The surface defects such as edges, corners, dopant atoms or heterogeneous domains usually serve as active sites in a photocatalytic reaction due to their particular electronic structure [29,30]. The selected area electron diffraction (SEAD) result (see the insert of Figure 2c) displayed discrete diffraction rings mixed with several diffraction spots. This means that the nanosheets are not homogeneous and monocrystalline. The inhomogeneity of the nanosheets' surface is conducive to the absorption and surface chemical reaction of organic molecules, and will be beneficial for photocatalytic activity.

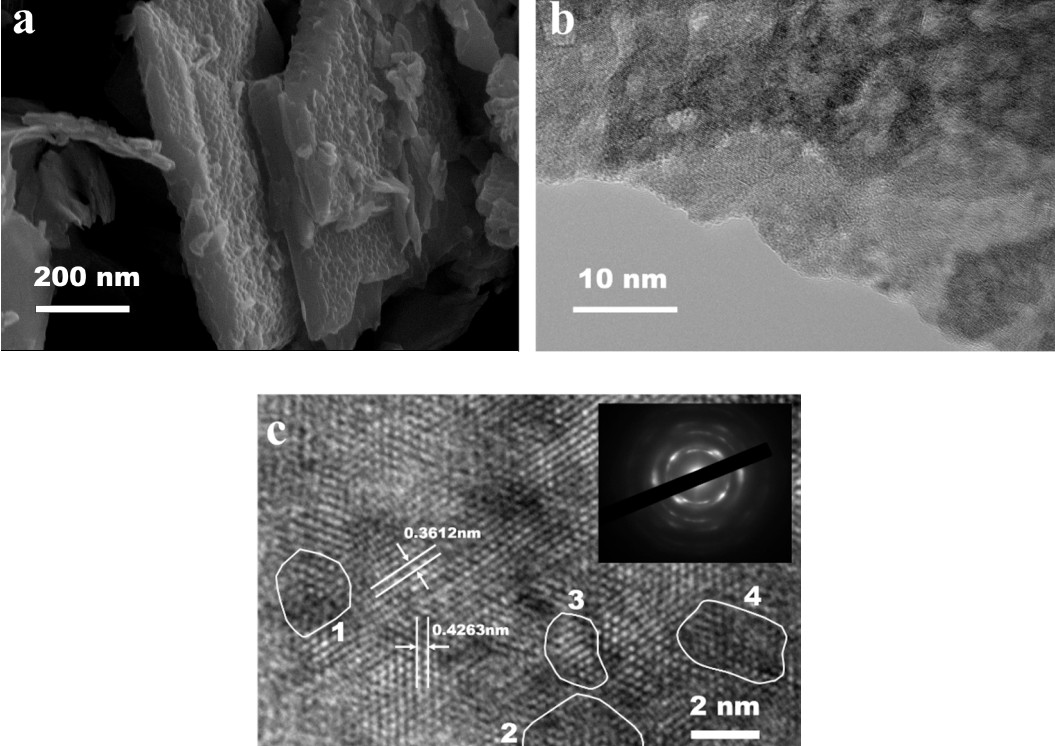

**Figure 2.** The morphology characterization of $NaBiS_2$. (**a**) SEM pattern; (**b**) low magnified TEM graph; (**c**) high magnified TEM graph, with nano-domains labelled.

The surface chemical states of the sample were checked by employing X-ray photoelectron spectroscopy. As shown in Figure 3, the XPS survey spectra reveal that the sample was composed of Bi, S, and Na elements. The peaks located at 290 eV and 530 eV correspond to C and O, which originate from the conducting resin and moisture adsorbed by the sample [23]. Two strong symmetrical peaks, located at 158.3 eV and 163.4 eV in Figure 3b, are attributed to $Bi4f_{7/2}$ and $Bi4f_{5/2}$ signals because of the spin-orbit splitting of Bi4f. Coincidentally, the S2p peak located at 164.03 eV, corresponding to elementary sulfur, is covered by the strong peak of $Bi4f_{5/2}$. Therefore, the S2p peak is hard to discern and accurately fit. Nevertheless, the S2s peaks can be fitted to 225.2 eV [31], though the intensity is weaker (Figure 3c). The XPS spectra reveal that sulfur exists in the form of $S^{2-}$. All of the specific peaks can be ascribed to the elements comprising $NaBiS_2$ properly.

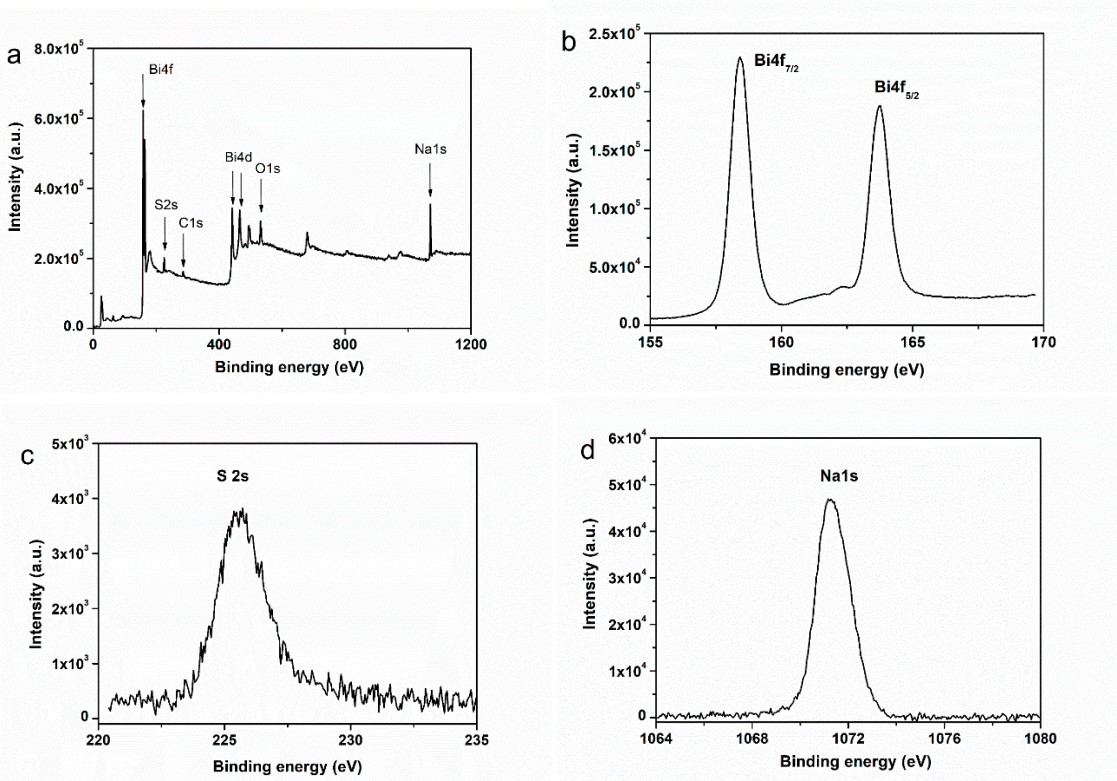

**Figure 3.** XPS spectra patterns of (**a**) full span, (**b**) Bi4f, (**c**) S2s, and (**d**) Na1s.

$NaBiS_2$ was previously reported as a promising visible-light-driven photocatalyst for rhodamine B(RhB) decomposition with the assistance of $H_2O_2$, and its bandgap was evaluated as 1.13 eV [24]. The modified augmented plane wave method (WIEN2k program) and HSE06 single-point calculations were employed to study the electronic energy structure of $NaBiS_2$. According to the difference of the theoretical model, the bandgap value varied between 1.12 eV and 1.37 eV [32,33]. The optical properties of $NaBiS_2$ were investigated first using diffuse-reflectance UV-Vis-NIR spectrophotometry. According to the diffuse reflection spectrum curve (Figure 4a), $NaBiS_2$ shows strong absorption from UV light (200 nm) to near-infrared light (about 1000 nm). The experimental bandgap of $NaBiS_2$ was estimated to be 1.32 eV, based on the absorption spectra according to the Kubelka-Munk theory, using the indirect bandgap semiconductor model. The details of DOS and band structures are shown in Figure 4b–d. The DFT calculation results reveal that the valence band consisted of Bi6p and S2p, indicating strong p–p hybridization between Bi6p and S2p, which originate from the bond of Bi and S in the crystal. The conduction band is mainly composed of S2p states, mixed with a few Bi6s states. In the conduction band, strong hybridization also appears between S2p and Bi6s, which results in the formation of the anti-bonding states. The energy band structure is shown in Figure 3c. The fermi

level was set as zero, shown by a dashed line. A partial enlarged view of the conduction band minimum (CBM) and valence band maximum (VBM) (Figure 4d) gives more detailed information. A slight mismatch was noted between the conduction band minimum and the valence band maximum. This implies that NaBiS$_2$ is an indirect bandgap semiconductor, and the indirect bandgap is defined to be 1.02 eV. The recombination of photo-induced carriers is a key factor that dramatically weakens the properties of a photocatalyst. The indirect nature of the bandgap was reported to be beneficial in suppressing the radiative recombination rate due to the momentum mismatch between the CBM and VBM [34]. Thus, the photocatalytic properties of NaBiS$_2$ are expectable. The calculated bandgap width value is smaller than the experimental result, which is possibly due to the well-known drawback of the DFT method.

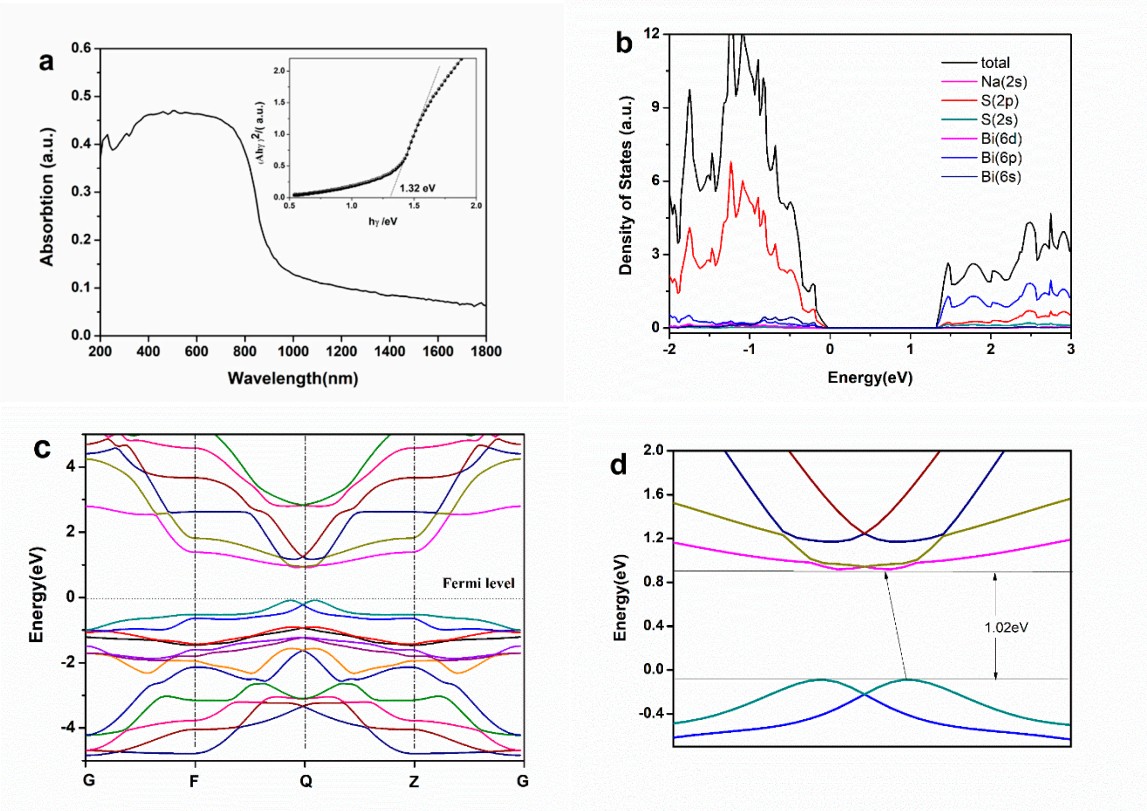

**Figure 4.** (**a**) UV-vis-NIR absorbance spectra of NaBiS2. Insert shows the bandgap estimated by the Kubelka–Munk method. (**b**) Density of states near the Fermi level and (**c,d**) the valence band maximum and the conduction band minimum.

The photocatalytic activity of NaBiS$_2$ was evaluated by monitoring the decoposition of methyl blue aqueous solution. A control experiment was carried out under the full spectrum of a xenon lamp without a photocatalyst, and negligible degradation of the methyl blue was observed after 120 min irradiation. It is reasonable to believe that methyl blue is stable in an aqueous solution even under light irradiation. Subsequently, degradation experimental measurements under UV (365 ± 5 nm) and visible light (420–800 nm) demonstrated the outstanding photocatalytic activity of NaBiS$_2$ (Figure 5a). As evaluated by the decrease in the UV–vis absorption spectra of an MB solution (Figure S1a–c), the degradation rates of MB are 72.1% and 73.3% when irradiated for 120 min with UV and visible light, respectively. Most notably, NaBiS$_2$ revealed impressive photocatalytic activity under NIR light. In total, 50.8% of the MB was degraded after 120min irradiation.

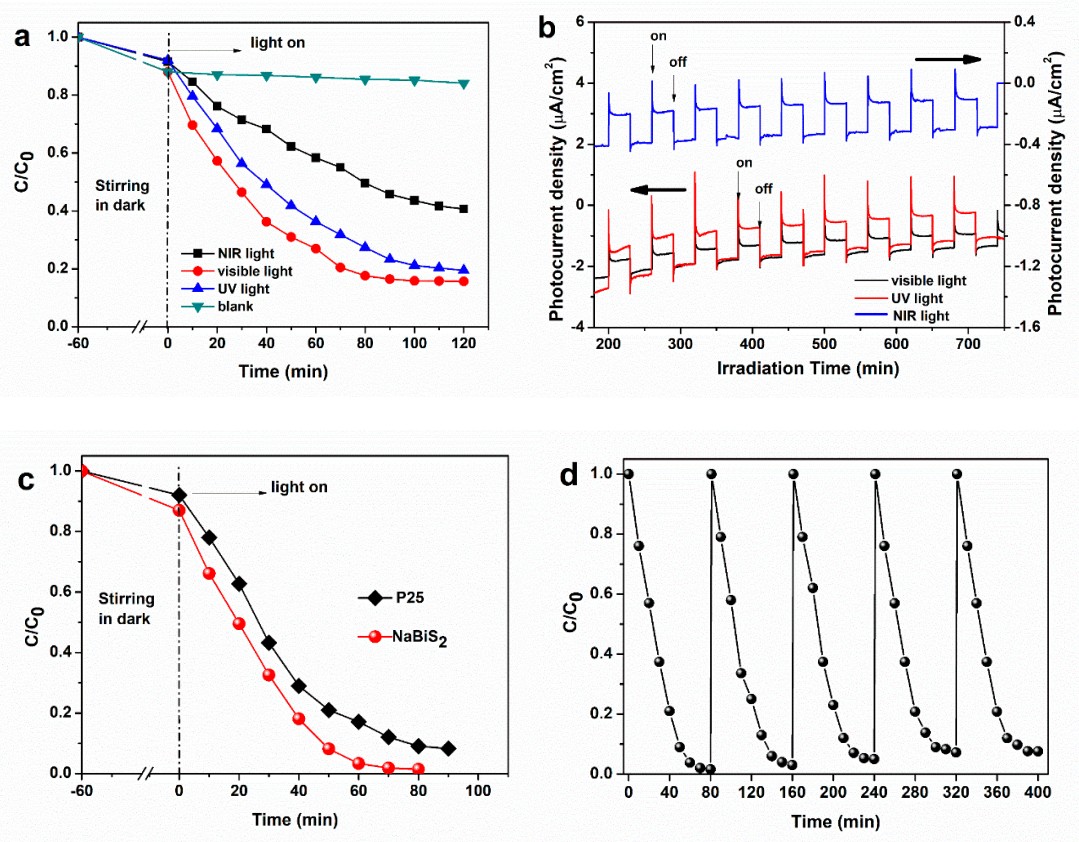

**Figure 5.** Photocatalytic activity and photocurrent measurement. (**a**) Degradation of MB with the irradiation of UV, visible and NIR light; (**b**) photocurrent density under UV, visible and NIR light; (**c**) degradation of MB with the full spectrum, compared with P25 without any other additive; (**d**) cycle runs under the full spectrum.

The broadband spectrum responsiveness performance was further assessed with photocurrent measurements. When $NaBiS_2$ modified FTO substrates were used as the working electrodes, a response photocurrent with repeatable on/off cycles upon light irradiation was observed. The photocurrent density curves under chopper light illumination are shown in Figure 5b. A sharp increase in photocurrent was recorded when the irradiation was turned on. Photocurrent is usually supposed to be relevant to the effective charge separation and charge transport over a photocatalyst's surface [35,36]. Various photocatalysts with photo-response activity display photocurrent response well. Photocurrent measurements indicate that, in $NaBiS_2$, electron–hole pairs generate at the moment of irradiation and can separate effectively, then transport to the surface where photocatalytic reaction occurs.

For comparison, methyl blue photocatalytic degradations by $NaBiS_2$ and P25 (Degussa, Germany) were carried out using a xenon lamp as a full spectrum light source (Figure 5c). $NaBiS_2$ demonstrates superior photocatalytic activity under unfiltered irradiation. The degradation rate of methyl blue reaches 99.6% after 80 min with the assistance of $NaBiS_2$, while P25 shows lower efficiency even after 90 min irradiation. Cycle runs under the full spectrum were also carried out to check the chemical stability of $NaBiS_2$. Though its photocatalytic performance deteriorates gradually, the degradation rate of MB still remains more than 91% after five cycle runs. The XRD result (shown in Figure S2) indicates that $NaBiS_2$ possesses good phase stability during a photocatalytic reaction. Negligible impurity was detected and the crystal structure of $NaBiS_2$ remained unchanged. Based on the above experimental and theoretical results, $NaBiS_2$ is confirmed as a novel stable photocatalyst with unexpected broadband spectrum photocatalytic activity.

## 3. Experimental Section

### 3.1. Preparation and Characterization of Photocatalyst

All the chemical reagents used in this work were analytical-grade reagents without any further purification. $Na_2S$ was dried in a vacuum oven at 120 °C for 8 h in advance. In a typical synthesis procedure of $NaBiS_2$, 10 mmol $Bi(NO)_3 \cdot H_2O$ was first dispersed into 40 mL deionized water and stirred for 10 min. NaOH (0.1 mol) and $Na_2S$ (0.1 mol) were dissolved in 20 mL deionized water. Then the above solutions were mixed and sealed in a Teflon-lined stainless steel autoclave, and treated at 160 °C under autogenous pressure for 12 h. After cooling to room temperature spontaneously, the sediment was separated and washed until the filtrate was neutral. After the sediment was dried in an electric oven, characterizations were conducted. In order to determine the optimal condition for $NaBiS_2$ synthesis, various amounts of NaOH and $Na_2S$ were employed during fabrication, comparatively following the above procedure.

The samples were analyzed by powder X-ray diffraction (XRD) with a Bruker D8-Advance diffractometer using monochromatized Cu $K\alpha$ ($\lambda$ = 1.54056 nm) radiation, with a scanning speed of 0.15°/s to check the phase firstly. The morphologies of the samples were observed by a field emission scanning electron microscope (JSM-7001F, JEOL) operating at a 5 kV accelerating voltage. UV-Vis-NIR diffuser reflectance (DRS) was conducted on a UV-Vis-NIR spectrometer (PerkinElmer, Lambda 950) to estimate the absorption spectra of $NaBiS_2$. The chemical states of the elements were measured by X-ray photoelectron spectroscopy (XPS) on a Thermo Fisher ESCALAB 250Xi instrument. The TEM images were recorded on a JEOL-JEM 2100 transmission electron microscope, using an accelerating voltage of 200 kV. The samples used for TEM observations were dispersed in ethanol, followed by ultrasonic vibration for 30 min, then a drop of the dispersion was placed onto a copper grid coated with a layer of amorphous carbon.

### 3.2. Photocatalytic Activity and Photochemical Measurements

The photocatalytic properties of the specimen were inspected by the photocatalytic degradation of methyl blue aqueous solution. Prior to the reaction, a 0.1 g sample was dispersed in 100 mL methyl blue solution (100 mg·$L^{-1}$) and stirred in the dark for 60 min to reach the adsorption-desorption equilibrium. A 10W LED with an emission wavelength of 365 ± 5 nm was used as the UV light. Visible light (420 nm < $\lambda$ < 800 nm) and NIR light ($\lambda$ > 800 nm) were obtained from a 300W xenon lamp with the assistance of cutoff filters. At regular time intervals, a certain amount of suspension was collected and centrifuged, and the residual supernatant was analyzed by UV-vis spectrophotometer (UV-3100, Hitachi). Finally, the photocatalytic efficiency was calculated by $C/C_0 \times 100\%$ (C is the final concentration and $C_0$ is the initial concentration of methyl blue). For stability measurements, the photocatalyst was centrifuged and collected after one photocatalysis cycle was completed, followed by washing and drying. Stability measurements were carried out under the same procedure as mentioned above.

Electrochemical measurements were performed on an electrochemical workstation (CHI 660, ChenHua, Shanghai) in 0.5mol·$L^{-1}$ $Na_2SO_4$ solution as the electrolyte. A standard three-electrode cell was used, employing a platinum wire as the counter electrode and a saturated calomel electrode in saturated KCl as the reference electrode. An $NaBiS_2$ modified FTO glass was used as the work electrode. It was fabricated as follows: 15 mg $NaBiS_2$ was first dispersed in 1 mL Nafion (1 wt%) solution and treated by ultrasonic scatter for 5 min. Then 0.5 mL solution was dropped on the FTO glass (1.5 cm × 1.5 cm) and dried in a vacuum oven to vaporize the solvent. During all of the measurements, the photocurrent was recorded at a bias of 0 V versus the reference electrode.

### 3.3. Theoretical Calculation

A theoretical investigation of the energy band of $NaBiS_2$ will be beneficial to the deep understanding of its properties. In this work, the band structures of $NaBiS_2$ crystal, including the project density of state (PDOS) and band width, were calculated. The geometry optimization and electronic properties

of crystalline $NaBiS_2$ were calculated under the density functional theory (DFT), as implemented in the Vienna ab-initio simulation package (VASP). The projector augmented wave (PAW) potentials were used as pseudopotentials to describe the interactions between the valence electrons and ions. The Perdew–Burke–Ernzerhof (PBE) functional [37] of generalized gradient approximation (GGA) was used to describe the exchange-correlation of the valence electrons. The computational procedures were carried out as follows: an optimization process was first performed using a conjugate gradient algorithm with a force tolerance of 0.02 eV/Å and a kinetic energy cutoff set at 500 eV. A Gamma k-point mesh of size $5 \times 5 \times 5$ was used for sampling the Brillouin zone for geometry optimization and the subsequent static calculations. The volume of the supercell was kept fixed until all forces were smaller than 0.01 eV/Å, corresponding to an energy convergence smaller than 0.0001 eV. This was small enough for the survey of the electronic structures. In total, 120 k-points were sampled along the band path in the Brillouin zone for the band diagram calculation.

## 4. Conclusions

In summary, $NaBiS_2$ nanosheets were synthesized facilely through a hydrothermal method at a low temperature, based on the experience of synthesis condition control. The alkaline amount is considered to play a key role in formation of $NaBiS_2$ and a threshold amount is needed. The as-synthesized polycrystalline $NaBiS_2$ shows multiple atomic configurations on the nanosheet's surface, including amorphous clusters and amorphous nano-domains. Its excellent photocatalytic activity for the degradation of MB in the range from UV to NIR light was confirmed, as well as its unexpected chemical stability under full spectrum irradiation. The band structure was theoretically investigated for the first time under the density functional theory, and $NaBiS_2$ is believed to be an indirect bandgap semiconductor. The present work gives the opportunity for the synthesis and application of $NaBiS_2$ as a broadband spectrum light-absorbing material.

**Supplementary Materials:** The following are available online at http://www.mdpi.com/2073-4344/10/4/413/s1: Figure S1: UV–vis absorption spectra curves of MB solution irradiated under UV, visible and NIR light, Figure S2: XRD patterns of $NaBiS_2$ before and after photocatalytic reaction.

**Author Contributions:** H.W. designed the project, conducted most of the experiments and wrote the manuscript; Z.X. performed part of the experiments and revised the manuscript with input from all the authors; X.W. and Y.J. took part in the discussion and revision of the manuscript. All authors have read and agreed to the published version of the manuscript.

**Funding:** This research was funded by National Natural Science Foundation of China, grant number 21875281.

**Acknowledgments:** This work was supported by National Natural Science Foundation of China (No. 21875281).

**Data Availability Statement:** The authors confirm that the data supporting the findings of this study are available within the article and its supplementary information.

**Conflicts of Interest:** The authors declare no competing financial interests.

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
