# Peer review of "NaBiS2 as a Novel Indirect Bandgap Full Spectrum Photocatalyst: Synthesis and Application"

_catalysts, doi:10.3390/catal10040413_

Round 1
Reviewer 1 Report
In the manuscript, the authors described the synthesis and broad range of photocatalytic properties of NaBiS2. These results will be informative for the researchers in the field of catalytic chemistry.
Whereas the reviewer thinks that the authors’ study in this manuscript is interesting, suggestive, and well-organized, some descriptions and discussions are not enough. The authors’ manuscript is not suitable for publication in “Catalysts” in the present form.
From these considerations, the reviewer recommends accepting for publication in " Catalysts," if the following issues are resolved.
- Page 2, line 54, "The band structure was investigated theoretically by DFT method for the first time,"; The authors should determine the scope and limitation of the words “for the first time” on the authors’ study of NaBiS2 by citing other DFT studies of NaBiS2 (derivatives).
- There is no discussion about the quantum efficiency of the photocatalytic reactions in each wavelength region. How do the authors evaluate the efficiency of NaBiS2 as a photocatalyst?
- Page 6 line 213~, “P25”; How did the authors prepare the “P25”? What are the basic properties of the authors’ “P25”?
- The Definitions of the following abbreviations should be provided in the vicinity of the words;
page 2 line 49, “NIR”,
page 2 line 55, “DFT”,
page 5 line 184, “RhB”,
page 6 line 213~, “P25.”
Author Response
Reply to referees` comments
Referee 1: Comments and Suggestions for Authors
In the manuscript, the authors described the synthesis and broad range of photocatalytic properties of NaBiS2. These results will be informative for the researchers in the field of catalytic chemistry. Whereas the reviewer thinks that the authors’ study in this manuscript is interesting, suggestive, and well-organized, some descriptions and discussions are not enough. The authors’ manuscript is not suitable for publication in “Catalysts” in the present form. From these considerations, the reviewer recommends accepting for publication in " Catalysts," if the following issues are resolved.
- Page 2, line 54, "The band structure was investigated theoretically by DFT method for the first time,"; The authors should determine the scope and limitation of the words “for the first time” on the authors’ study of NaBiS2 by citing other DFT studies of NaBiS2 (derivatives).
Reply: After carefully surveying and reviewing the relative researches about alkali metal bismuth chalcogenide, we re-edit the sentence and remove the statements such as “for the first time” for the sake of rigor. Please see Line 54 on Page 2, Line 102 on Page 3 and Line 186-188 on Page 6. And other DFT studies of NaBiS2 (derivatives) are cited. Please see Line 178-181 on Page 6.
- There is no discussion about the quantum efficiency of the photocatalytic reactions in each wavelength region. How do the authors evaluate the efficiency of NaBiS2 as a photocatalyst?
Reply: Thanks for the comment on our negligence of efficiency evaluation definition in our manuscript. The efficiency of NaBiS2, that means the photocatalytic performance of NaBiS2 is evaluated by the degradation of methyl blue as model pollutant. The definition of photocatalytic efficiency was presented in revised manuscript for better understanding. Please see Line 88-89 on Page 3.
As for photocatalytic application theme, utilization of model pollutant especially dye molecule is the common method evaluating the performance and the possibility for practical application. Due to the lack of understanding about the molecular mechanisms and electron transport, the method for quantum efficiency evaluation of photocatalytic degradation reactions has not been reported yet to our knowledge. We think it will not bewilder the audiences who is interest and expert in this topic. In view of this, we didn`t discussion the quantum efficiency of the photocatalytic reactions.
- Page 6 line 213~, “P25”; How did the authors prepare the “P25”? What are the basic properties of the authors’ “P25”?
Reply: Thanks for the kind remind from referee. P25 is the commercial name of TiO2 nanoparticle produced by Degussa (Germany) with the CAS NO. 13463-67-7. It has the anatase/rutile ratio at about 80/20, and is usually employed as the certified reference materials in photocatalytic degradation research due to its outstanding performance under UV light and its good repeatability, such as reported by Yu (Applied Catalysis B: Environmental, 2016) and Han (Chemistry A European Journal, 2017). The specific surface area of commercially available P25 is about 50m2/g and the average particle size is 20nm. The P25 used in our work is purchased from Sinopharm (Beijing, China) and used without any further treatment. We employed titanium oxides(P25) in order to illustrate the photocatalytic activity of NaBiS2 under simulated sunlight illumination and convenient for comparison to other full spectrum active photocatalysts. In consideration of these, we did not present the basic properties and the preparation of P25 in our manuscript, and we think it will not bewilder the audiences who is interest and expert in this topic.
- The Definitions of the following abbreviations should be provided in the vicinity of the words;
page 2 line 49, “NIR”,
page 2 line 55, “DFT”,
page 5 line 184, “RhB”,
page 6 line 213~, “P25.”
Reply: In accordance with the reviewer’s helpful comment, we rearrange the above-mentioned abbreviations in the manuscript. The definitions are presented before the abbreviations are used. Please see Line 43 and Line 54 in Page 2, and Line 177 in Page 6. We list the manufacturer and origin information of P25 instead of definition because it is a commercial name of nano-titanium oxides with the CAS NO. 13463-67-7. Please see Line 232 in Page 8.

Reviewer 2 Report
I found only a few minor errors that need to be corrected.
- Line 77: there is „wer” and should be „were”
- Line 63: there is another font in sentence.
- Sentence 65: The sentence couldn’t start with numbers. I think that this sentence must be corrected.
Author Response
Reply to referees` comments
Referee 2: Comments and Suggestions for Authors
I found only a few minor errors that need to be corrected.
- Line 77: there is „wer” and should be „were”
- Line 63: there is another font in sentence.
- Sentence 65: The sentence couldn’t start with numbers. I think that this sentence must be corrected.
Reply: Thanks for the critically review on our manuscript. In accordance with the respectable reviewer’s helpful comment, these sentences in the manuscript have been edited. The font and the misspelling are corrected. Please see Line 62, Line 64 and Line 76 on Page 2.
